# Associations between Feminine Gender Norms and Cyber Dating Abuse in Female Adults

**DOI:** 10.3390/bs9040035

**Published:** 2019-03-29

**Authors:** Beatriz Villora, Santiago Yubero, Raúl Navarro

**Affiliations:** Department of Psychology, Faculty of Education and Humanities, University of Castilla-La Mancha, 16071 Cuenca, Spain; santiago.yubero@uclm.es

**Keywords:** Cyber dating abuse, femininity, gender roles, adults

## Abstract

Gender norms and the co-occurrence of perpetration and victimization behaviors have been examined as key factors of female dating violence in offline contexts. However, these relationships have not been analyzed in digital environments. This is why the present study had a twofold objective: (1) exploring the co-occurrence nature of cyber dating abuse by examining to what extent victimization and perpetration overlap; (2) examining the associations between conformity to feminine gender norms and cyber dating abuse among female adults who are perpetrators or victims. The sample study included 1041 female university students (mean age = 20.51) from central Spain. The results indicated that 35.8% of the sample reported being a victim and a perpetrator of cyber dating abuse at the same time. Indeed, the hierarchical regression analyses revealed a close association between perpetration and victimization behaviors in both the direct and control forms of abuse examined. Our analyses did not reveal any significant associations between conformity to female gender norms and perpetration or victimization for any cyber dating abuse form examined. Our results are discussed in the light of previous research and after considering limitations, practical implications and future research directions.

## 1. Introduction

Early adolescence is a time when youths begin to explore their sexuality and engage in romantic relationships. In a national study including 14,956 students in grades 9–12, 68.3% stated having been in a relationship in the last year [1]. More importance is attached to romantic relationships with time when they become more stable [2]. In young adulthood, romantic relationships are a common aspect of healthy development [3]. However, young adults aged 20–24 years are at higher dating violence risk, and this period coincides with university life [4]. An international study conducted with university students from 32 nations revealed that approximately one third of the surveyed females and males were victims of their partner’s abuse in the previous 12 months [5] 

In the last decade, abuse within romantic relationships has been acknowledged as a relevant social problem [6]. Nonetheless, the literature has not paid the same attention to the aggression exercised in the relationships formed by young adults to those formed by adults [7]. Dating violence includes any act of physical, sexual, stalking, psychological or emotional aggression [8]. Research has traditionally focused on the physical aggression that men display with women, but it is now well known that both men and women can be perpetrators and victims, and that different forms of abuse exist in romantic relationships [9]. 

Previous research has revealed how certain forms of abuse in romantic relationships are perpetrated via technology [10]. Digital tools such as email, messenger applications and social networks, which now form a fundamental part of young people’s lives, can be used unhealthily [11], and provide instruments with which to harass, control, abuse and stalk partners or former partners [12]. The aggression caused by these tools is called cyber dating abuse, and includes both control abuse (theft, undue use of passwords, spreading compromising or secret information) and direct aggression-type (threats and insults in private/public) behaviors [13].

Several studies have found that cyber dating abuse is associated with offline experiences of psychological, physical and sexual aggression perpetrated by one’s partner [14,15,16,17]. A high prevalence of around 50% has been documented for perpetration and victimization at university [18,19], although the factors associated with online abuse as part of romantic relationships are barely known [20]. 

### 1.1. Dating Abuse as a Co-Occurring Behavior

Research into offline forms of abuse has found that dating victimization is positively correlated with perpetration in both males and females [9,21]. Indeed, having been a victim of one’s partner’s abuse has been found to be one of the strongest predictors of female perpetration [22]. Likewise, other studies have reported that when women abuse their partners, these women also tend to suffer greater victimization by their partners [23]. Along the same line, being a victim of cyber dating abuse in digital environments seems to be a strong predictor of female cyber dating abuse perpetration, and vice versa. In other words, both members of the romantic relationship may act as perpetrators and victims [17,24,25,26,27]. 

The overlapping nature of offline and online dating abuse can be explained by several reasons [28]: (1) the intergenerational transmission of violence based on the Social Learning Theory [29] (this theoretical framework proposes that an individual learns behaviors through his/her social interactions). The presence of violent situations in families is outlined as a risk factor for youths to repeat similar strategies of violence in their relationships with their partners. Lewis, Travea and Fremouw [30] found that females who said they were victims and perpetrators of abuse were significantly more likely to have witnessed parental intimate partner violence. (2) In line with the Social Learning Theory, violence breeds violence. Young people who have friends who use dating violence are at increased risk of using dating violence [31]. Palmetto, Davidson, Breitbart and Rickert [32] found that victims of violence are more likely to use similar strategies with perpetrators, or may relate with individuals who are inclined to use abuse in their romantic relationships. (3) Self-defense. (4) Revenge, deteriorated relationship, jealousy and control. 

### 1.2. Gender and Dating Abuse Behaviors

Many studies have analyzed the differences involved in cyber dating abuse according to participants’ gender, but their results are contradictory in both perpetration and victimization forms [14]. Some studies have found no significant differences between genders [33], while others report higher levels of perpetration [34] and victimization [35] for men. Conversely, other studies have reported higher perpetration and victimization rates for women [18,36]. 

Several researchers have pointed out the importance of going beyond analyzing only gender differences and considering which gender-related socio-cultural factors might explain the possible differences between men and women [37,38]. While sex is considered a biological category (genital, brain, hormonal, etc.), gender is understood to be the result of a development process in which social norms and expectations are internalized [39]. Following the Gender Role Theory [40], gender is a social construction and, therefore, research should examine differences in cyber dating abuse according to the adherence to traditional gender roles that set different behavior patterns for males and females [41]. Gender roles are defined as “those shared expectations about appropriate conduct that apply to individuals solely on the basis of their socially identified sex” [42]. According to the Gender Role Theory, men would be socialized to demonstrate their authority and control and would, therefore, be more inclined to use control behaviors, including abuse. Conversely, women would be socialized to be passive and complaisant, which are traits with which abuse seems less compatible [43]. 

Bearing this in mind, conformity to feminine norms refers to adhering to traditional social standards about femininity that affect the way women express themselves, think and behave, whereas non-conformity is understood as not adhering to social expectations about what conventional femininity would involve [44]. Women who adhere to feminine norms may be rewarded by meeting social expectations, while those who do not conform to feminine norms might be treated with contempt, or may even become victims of various forms of aggression [45].

Socio-cultural influences shape the gender role expectations and standards that constitute gender role norms [44,46]. Consequently, the degree to which gender roles are emphasized may vary across cultures [47]. Sánchez-López and Cuéllar-Flores [48] conducted research to assess whether young Spanish university students’ sense of femininity was similar to American college students’ sense of femininity. They reported that the femininity gender role expectations among Spanish women were similar to the gender role expectations among North American women. However, the study also revealed that Spanish female students and North American female students reported differences in the degree of conformity to gender roles. For example, Spanish women obtained higher scores on gender roles related to home-loving, but lower scores on gender roles related to interest in romantic relationships, care of children, sexual fidelity and thinness. No differences were found in the gender roles related with modesty and appearance.

Although the academic literature comprises lots of research data on the association between conventional gender beliefs and men perpetrating violence, very little research has been conducted about this relation and female violence in romantic relationships [49]. However, research has found that conformity to gender stereotypes and traditional gender roles is positively associated with perpetrating offline aggression forms among women [9]. For example, in a university student sample of heterosexual couples, Burke, Stets and Pirog-Good [50] reported that a more traditional feminine identity was associated with women perpetrating physical and sexual abuse. Nonetheless, the opposite results have also been reported by studies that have analyzed bullying behaviors, which have found that traditional femininity was negatively associated with direct aggression forms [38,51]. 

In digital environments, research has also reported that traditional feminine personality traits are associated with aggressive behaviors. For example, Wright [52] found that adolescents showing more feminine traits participated more in cyber aggression forms via mobile phones and social network sites with higher cyber relational aggression rates. Regarding online abuse in romantic relationships, the role played by gender roles has barely been examined. In adolescence, research has shown that if female adolescents conform to gender stereotypes, they could be more significantly related with perpetrating control and digital stalking conduct [10,49]. 

Research into victimization processes has shown that traditional gender roles may increase the risk of violence against women. Conforming to certain conventional gender beliefs (e.g., maintaining relationships by attaching a maximum value) might indirectly increase the risk of suffering victimization [53]. Other authors, however, suggest that women’s victimization is the result of not conforming to a socially assigned gender role [54].

Different studies conducted with samples of female university students have reported that holding a traditional feminine identity and conforming to the traditional values of gender roles are associated with victimization by physical and sexual abuse [50,55,56]. Nonetheless, other researchers have found no significant association between victimization in different forms of aggression (physically forced, verbally coerced sexual and bullying) and adhering to traditional femininity norms among women [28,57].

Gender roles have also been associated with victimization via electronic forms of aggression. Previous research has shown that women who conform to traditional gender stereotypes are less likely to report victimization by cyber stalking [58]. However, despite previous studies having revealed that women are more likely to be victims of cyber dating abuse [18,36], researchers have not yet examined how gender roles are related with the victimization probability in this abuse type.

### 1.3. The Present Study

Based on the aforementioned studies, the objectives of the present work were to: (1)Analyze the prevalence rates of cyber dating perpetration and victimization in a sample of female university students. Female university students were expected to report different victimization and perpetration experiences and the most frequent form of abuse would be control abuse (H1).(2)Explore the co-occurrence of cyber dating abuse by analyzing to what extent victimization and perpetration overlap. Given previous research on online and offline contexts, a correlation was hypothesized between cyber dating victimization and perpetration in both the control and direct forms of abuse.(3)Examine the differences in conformity to feminine gender norms among those women who claim to be perpetrators or victims of cyber dating abuse. Considering previous research that has analyzed the associations between gender role norms and different forms of aggression, females showing higher conformity to feminine gender norms were expected to be more involved as perpetrators or victims of cyber dating abuse.

## 2. Method

### 2.1. Participants

We performed cross-sectional analyses with the data that we obtained between February and March 2018 from 1086 undergraduate women studying at a Spanish university (with approximately 23,000 students) in central Spain. Before the analysis, all the data were checked for missing values. Missing data were not statistically imputed. Participants were required to provide valid data for all the study variables. Four cases were excluded as some data were missing in certain measures. Thus, the final number of participants with full records for the variables included for the analysis was 1082. A data analysis was done with the participants reporting having been in a romantic relationship in the past 12 months or presently being in such a relationship. Forty-one cases were excluded for that reason. The final sample included 1041 undergraduate women, whose age range was 18–41 years (*M* = 20.51; *SD* = 3.00); 57.3% were studying social sciences, 7.7% applied sciences, and 35% physical and health sciences.

### 2.2. Measurement Variables and Instruments

The participating female university students self-reported information on demographic variables: their ages and sexual orientations. The instruments shown below were used to analyze the study variables.

*Cyber dating abuse.* The Cyber Dating Abuse (CDA) Questionnaire [24] includes 20 items about various types of CDA, including different online abusive behaviors ranging from perpetration to victimization perspectives. The CDA questionnaire comprises two factors: direct abuse (e.g., aggressive action for the deliberate intention of hurting one’s partner/former partner). An example of an item is: “I threatened my partner/former partner using new technologies to physically hurt her/him”; control abuse, referring to the use of electronic means to control one’s partner/former partner. An example of an item is: “By mobile applications, I controlled the time that my partner/former partner last connected”. All the CDA items are scored on a 6-point scale as follows: 1 (never); 2 (not in the last year, but before); 3 (once or twice); 4 (3–10 times); 5 (10–20 times); 6 (more than 20 times). In our study sample, the perpetration scale reliability was 0.84 for control perpetration and 0.78 for direct perpetration. The victimization scale reliability was 0.82 for control victimization and 0.76 for direct victimization.

*Feminine Gender Norms*. In order to evaluate participants’ conformity to feminine gender norms, the abbreviated version of the Conformity to Femininity Norms Inventory (CFNI) was employed [44,45], which was adapted to Spanish by [59]. The CFNI contains 45 items, which are all answered on a 4-point scale (0-strongly disagree to 3-strongly agree) to evaluate conformity to a range of femininity norms from the US society. Feminine gender norms are grouped into nine scales: (1) Relational, which refers to develop friendly and supportive relationships with others (item example: “I believe that my friendships should be maintained at all costs”); (2) Thinness, related to pursue a thin body ideal (item example: I would be happier if I was thinner”); (3) Modesty, referring to refrains from calling attention to one’s talents or abilities (item example: I always downplay my achievements”); (4) Domestic, related to home maintenance (item example: It is important to keep your living space clean”; (5) Romantic relationship, which refers to invest self in romantic relationships (item example: “Having a romantic relationship is essential in life”); (6) Invest in appearance, which is related to commit resources to maintain and improve one’s physical appearance (item example: I spend more than 30 min a day doing my hair and makeup); (7) Sexual fidelity, which refers to keep sexual intimacy contained to one committed relationship (item example: I would feel guilty if I had a one-night stand”); (8) Care for children, which is related to take care of and spend time with children (item example: “Taking care of children is extremely fulfilling”); (9) Sweet and nice, referring to being nice with known and unknown people (item example: “I always try to make people feel special”). The results obtained in Spain support the suitability of the CFNI as a multidimensional gender measure to be used in this country [39,48]. Reliability in the current sample yielded a Cronbach’s alpha coefficient of 0.76 for the total scale, 0.74 for the relational subscale, 0.76 for the thinness subscale, 0.80 for the modesty subscale, 0.79 for the domestic subscale, 0.74 for romantic relationships, 0.83 for the appearance subscale, 0.86 for the sexual fidelity subscale, 0.76 for care for the children subscale and 0.83 for sweet and nice. 

### 2.3. Procedure

Self-reported group class-administered pencil-and-paper questionnaires were employed to collect data. One researcher handed out the questionnaires to those participating, explained the meaning of some items, and answered questions if any were asked. Participants were ensured that their answers would remain anonymous and they could withdraw from the study whenever they wished to. The procedure took place in each group class and lasted approximately 15 min. The procedure to collect data was followed during usual classroom schedules, which lasted 8 weeks. The study complied with Spain’s legal requirements. The Clinical Research Ethics Committee of Virgen de la Luz Hospital in Cuenca approved the study protocol and all the subjects signed informed consent forms prior to participation in the study.

### 2.4. Analysis Plan

Details of the general descriptive of the independent variables were provided first. Then, the descriptive data related to participants’ involvement in cyber dating abuse were examined. Participants’ categorization as victims, perpetrators or perpetrators-victims in each form of abuse was made by following a criterion used by previous cyber dating researchers [24]. The participants who indicated suffering, but not perpetrating, once or more in at least three of the abusive behaviors included in the questionnaire were classified as victims. The participants who reported perpetrating, but not suffering, once or more in at least three of the abusive behaviors were classified as perpetrators. The participants who indicated suffering and perpetrating once or more in at least three of the abusive behaviors were classified as perpetrators-victims. The remaining students were considered to not be involved in cyber dating abuse. In order to assess the relationship linking cyber dating victimization and perpetration in the last 12 months, victimization and perpetration behaviors were correlated on the control and direct subscales and the total score (the sum of both the control and direct abuses). Hierarchical regressions were used to ascertain whether the nine subscales of the feminine gender norms were related with cyber dating abuse (by analyzing both the control and direct forms of abuse) both over and above the experience of being a perpetrator/victim of cyber dating abuse. Data were analyzed using Version 24.00 of SPSS.

## 3. Results

### 3.1. General Descriptive and Prevalence Rates of Cyber Dating Abuse

Table 1 shows the means and standard deviations of all the study variables for the whole sample. The prevalence rates of cyber dating abuse perpetrators/victims are found in Table 2. Whereas nearly 50% of our sample did not report being a victim/perpetrator of cyber dating abuse in their relationships, 35.8% indicated control and direct abuse as being co-occurring (both receiving and perpetrating abuse behaviors). On average, female students reported having perpetrated slightly fewer acts of abuse (*M* = 1.36, *SD* = 0.45) than being victims of abuse in the last 12 months (*M* = 1.42, *SD* = 0.63), *t* (1041) = −3.20, *p* < 0.001. Pearson’s correlations were conducted for further analyses. The correlation observed between overall cyber dating abuse victimization and overall cyber dating abuse perpetration was 0.516 (*p* < 0.001). The correlation between perpetrating direct abuse and being a victim of direct abuse was 0.403 (*p* < 0.001), while that between perpetration and victimization of control abuse was 0.535 (*p* < 0.001). 

### 3.2. Associations between Femininity Norms Inventory (CFNI) Scales and Cyber Dating Abuse Victimization

Hierarchical regression revealed that being heterosexual and perpetrating direct cyber dating abuse accounted for 16% of the variance observed in the scores for being victimized by direct cyber dating abuse. After accounting for age, sexual orientation and direct perpetration, a significant association was found with direct cyber dating victimization for only one CFNI scale, but it was very weak and accounted for only an additional 1% of variance in direct victimization (see Table 3). More self-investment in a romantic relationship was associated with higher direct victimization levels. 

Hierarchical regression also revealed that perpetrating control cyber dating abuse accounted for 29% of the variance in the scores on being a control cyber dating abuse victim (*B* = 0.635, *SE* = 0.031, *p* < 0.001). After accounting for control perpetration however, the CFNI scales were not significantly related to control cyber dating victimization (r^2^ change: 0.003, *p* = 0.23).

### 3.3. Associations between the Femininity Norms Inventory (CFNI) Scales and Cyber Dating Perpetration

Hierarchical regression demonstrated that being a direct cyber dating abuse victim explained 16% of the variance in the scores on perpetrating direct cyber dating abuse (*B* = 0.229, *SE* = 0.016, *p* < 0.001). After accounting for received victimization, however, the CFNI scales did not significantly relate with perpetrating direct cyber dating abuse (r^2^ change: 0.006, *p* = 0.63).

Hierarchical regression also indicated that being homosexual and a control cyber dating abuse victim explained 29% of the variance in the scores on perpetrating control cyber dating abuse. After accounting for age, sexual orientation and control victimization, only two CFNI scales were significantly related to control cyber dating perpetration, but they were very weak and explained only an additional 1% of the variance in the control cyber dating abuse (see Table 4). Greater sexual fidelity was related to higher control perpetration levels, while less supportive and friendly relationships with others were associated with higher control perpetration levels.

## 4. Discussion

The objective of the present study was to extend basic empirical knowledge about cyber dating abuse by exploring victimization-perpetration co-occurrence and examining the differences in conformity to feminine gender norms between victims and perpetrators in a sample of female university students.

Our results corroborated our first hypothesis. We found that 35.8% of women indicated being victims/perpetrators of cyber dating abuse as opposed to the 8% who reported being only victims and 13.6% being only perpetrators. The regression analysis also indicated a moderate to strong association between perpetration and victimization due to cyber dating abuse. These findings support previous research into perpetration and victimization co-occurring in both offline and online forms of abuse [17,25,60,61], which indicates that this abuse type tends to have an overlapping nature. In other words, female victims are more likely to be perpetrators, and female perpetrators are more likely to become victims [13]. This result suggests that mutual violence patterns exist in the romantic relationships formed by university students, and when a partner received some abuse type, her response may be defensive, which is also violent in nature [4]. These results can be explained following the Social Learning Theory, according to which behaviors are learned through social interactions. The co-occurrence of victimization and perpetration could be explained as the mutual learning of abusive behavior between partners. In line with this, Palmetto, Davidson, Breitbart and Rickert [32] found that women reporting being victims and perpetrators of abuse also reported higher victimization rates than women indicating being only victims. They also reported higher perpetration rates compared to women indicating only being perpetrators. However, our cross-sectional design did not allow us to identify whether female abuse led to male abuse, or if female abuse was used as a means of self-defense against male abuse. Future studies should attempt to understand: (1) why women get involved in cyber dating abuse (e.g., self-protection, provocation, jealousy, etc.) as previous research has shown that abuse motivation varies between men and women [24]; (2) how the co-occurrence dynamic operates in this abuse type; (3) what risk factors increase the likelihood of engaging in both the perpetrator and victim roles. Our results could also be related with the family environment and peer relationships. Lewis, Travea and Fremouw [30] found that the women who had been victims and perpetrators were significantly more likely to have witnessed parental intimate partner violence. Likewise, research has documented that young people with friends who use dating violence are at increased risk of using dating violence [31]. Future research should analyze these potential relationships.

Regarding conformity to feminine gender norms, we hypothesized that there would be differences in conformity to feminine gender norms among women who perpetrate or are victims of cyber dating abuse. However, our results did not support this hypothesis because no significant associations were found between conformity to feminine gender norms and perpetrating direct and control abuse. We found only a significant association linking three feminine gender norms and control perpetration and direct victimization. However, the statistical weakness of these relationships did not allow us to suggest that any significant relation existed.

There may be several explanations for these results. First, the young female students in our sample obtained low scores for conformity to female gender role norms, which might suggest a shift from traditional gender roles among female university students brought about by the major socio-cultural changes that Spain has witnessed and the recent struggle for women’s basic rights in this country [48]. However, female university students’ responses could also be biased by social desirability. 

Another factor that could well contribute to our results is that some youths do not always identify cyber dating abuse behaviors as being a form of abuse [62]. Abusive control/intimidation behaviors are sometimes justified by youths as not being “very important” or form part of the normal interaction involved in romantic relationships. Thus, youths tend to trivialize abusive behaviors as “a joke”. It has also been found that females tend to normalize frequent digital control behaviors more than males [63].

In line with our results, some studies have found no associations between gender stereotypes and perpetrating cyber stalking for university students [58], or significant relationships between femininity and indirect aggression in both the victimization and perpetration forms [64]. Similarly, Zuo et al. [56] did not report any association between gender role attitudes and experiencing verbal sexual coercion via technology. These results suggest that the indirect nature of abuse conducted via information and communication technologies (ICT) may influence how gender norms operate, and the context in which aggression is exercised might moderate the impact of these norms. 

Finally, another explanation could lie in other risk factors that have not been assessed herein, but could moderate the influence of gender norms on exercising abusive behaviors, such as violence acceptance beliefs. Indeed, a longitudinal study has revealed that conformity to traditional gender roles was not associated with perpetrating dating abuse among those adolescents who reported low levels of violence acceptance beliefs [65]. Further research is necessary to shed some light on these findings.

### Limitations and Directions for Future Research

This article has some limitations that should be taken into account. First, our results were obtained by self-report measures. Despite such measures being widely used to assess cyber dating behaviors [43], we must bear in mind respondents’ honesty when they answer questions on such delicate matters because social desirability might affect them. Second, future research must contemplate acquiring further data using reports obtained from both partners about their abusive behaviors, and by employing other methods, such as qualitative interviews or focus groups, to gain a complete picture of the cyber dating abuse phenomenon. Third, the generalization of the present results is limited because the study sample comprised only female university students from a given area in central Spain and was mainly heterosexual. Futures studies should replicate these findings with other different cross-cultural, adolescents and more diverse samples in terms of sexual orientation and gender identities. Fourth, the study’s cross-sectional design restricts its capability to make causal inferences. Future longitudinal studies need to be conducted to examine the causal relation linking cyber dating abuse, gender norms and the reciprocal nature of aggression. Fifth, we did not differentiate between offensive and defensive forms of perpetration, which would prevent inferences being made about the reciprocal nature of abuse [66]. Sixth, although our analysis of the association between conformity to feminine norms and cyber dating abuse analyzed the victimization and perpetration co-occurrence, assessing other control variables would be worthwhile because their inclusion would enable us to better know the association linking gender norms and abusive behaviors [67]. Finally, it is worth stressing that our study reported high online cyber dating abuse rates in romantic relationships, but this did not indicate that abuse between men and women was the same. The meaning and reasons for men and women differ [68] and, therefore, future research must bear in mind such aspects.

Despite these limitations, our results have special practical implications as they indicate that intervention/prevention programs must address efforts to both men and women because, as our findings reveal, cyber dating abuse is a generalized practice among female university students. Moreover, the co-occurrence nature that cyber dating abuse takes informs us that women are not only victims, but also perpetrators. So, prevention programs must stress the factors that favor such reciprocity [13] by raising more awareness among male/female youths about how their conduct affects their partners and the processes that lead them to display such conduct [69].

## Figures and Tables

**Table 1 behavsci-09-00035-t001:** Summary statistics of the study variables.

Measures	Mean	SD	Range
Age	20.51	3.00	18–41
Sexual Orientation 89.5% Heterosexual			
Direct perpetration	1.12	0.29	(1–4)
Control perpetration	5.20	1.61	(1–5)
Direct victimization	1.22	0.52	(1–6)
Control victimization	1.66	0.88	(1–6)
**Femininity norms**			
Relational	11.07	2.25	(2–15)
Care for children	9.51	3.75	(0–15)
Thinness	6.38	3.89	(0–15)
Sexual fidelity	4.51	3.43	(0–15)
Modesty	6.16	2.40	(0–15)
Romantic relationship	6.16	2.90	(0–15)
Domestic	11.41	2.51	(2–15)
Invest in appearance	9.03	3.12	(0–15)
Sweet and nice	10.61	2.36	(3–15)

**Table 2 behavsci-09-00035-t002:** Prevalence rates of cyber dating abuse.

	Direction of abuse			
	None	Victimization and Perpetration	Victimization Only	Perpetration Only
Forms of abuse	n (%)	n (%)	n (%)	n (%)
Direct abuse	808 (77.5)	73 (7)	92 (8.8)	70 (6.7)
Control abuse	474 (45.4)	348 (33.4)	86 (8.2)	135 (12.9)
Total (direct and/or control abuse)	445 (42.7)	373 (35.8)	83 (8)	142 (13.6)

Note: n, number of participants in each category; %, percentage of participants in each category.

**Table 3 behavsci-09-00035-t003:** Hierarchical regression analysis analyzing the association between feminine norms and direct cyber dating victimization.

Variable	B	SEB	β	R^2^ Change
Step I				
Direct perpetration	0.703	0.050	0.401 **	0.163
Age	0.003	0.005	0.016	
Sexual orientation	0.095	0.048	0.056 *	
Step II				
Feminine norms subscales				0.010
Relational	−0.004	0.007	−0.017	
Care for children	0.003	0.006	0.015	
Thinness	0.004	0.004	0.033	
Sexual fidelity	0.001	0.005	0.004	
Modesty	−0.010	0.006	−0.048	
Romantic relationship	0.009	0.004	0.063 *	
Domestic	−0.008	0.006	−0.038	
Invest in appearance	−0.008	0.005	−0.050	
Sweet and nice	0.000	0.007	0.002	

*B*, coefficient; SEB, standard error; β, odds ratio; R^2^, r-squared; * *p* < 0.01; ** *p* < 0.001.

**Table 4 behavsci-09-00035-t004:** Hierarchical regression analysis analyzing the association between feminine norms and control cyber dating perpetration.

Variable	B	SEB	β	R^2^ Change
Step I				
Control victimization	0.455	0.022	0.536 ***	0.291
Age	−0.004	0.007	−0.015	
Sexual orientation	−0.201	0.064	−0.082 **	
Step II				
Feminine norms subscales				0.010
Relational	−0.020	0.010	−0.059 *	
Care for children	0.003	0.006	0.015	
Thinness	0.007	0.005	0.038	
Sexual fidelity	0.013	0.006	0.061 *	
Modesty	−0.014	0.008	−0.044	
Romantic relationship	−0.004	0.008	−0.015	
Domestic	0.002	0.008	0.006	
Invest in appearance	0.001	0.007	0.002	
Sweet and nice	0.013	0.010	0.039	

*B*, coefficient; SEB, standard error; β, odds ratio; R^2^, r-squared; * *p* < 0.05; ** *p* < 0.01; *** *p* < 0.001.

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
