# Peer review of "Associations between Feminine Gender Norms and Cyber Dating Abuse in Female Adults"

_behavsci, 2019, doi:10.3390/bs9040035_

Round 1

Reviewer 1 Report

The article is very interesting and concise. It deepens in a topic of interest and understudied such as that of female violence in romantic relationship and  analyzing the socio-cultural factors related to gender is a great addition to the literature.

Before accepting it for publication I would like the authors to address some minor concerns. Please do not be scared by the amount of points (13) I am making, as all of them are mostly minor remarks and it is my hope that they will help to improve the manuscript

Abstract

Point 1. In the abstract the authors state

 Indeed, the  hierarchical regression analyses revealed a close association between perpetration and victimization  behaviors in both direct and control forms of abuse examined.

However I am not sure if both forms of perpetration and victimization were included in the analysis, please see the point 11 in the results section.

Point 2. I find myself wondering, does the first aim "explore the bidirectionality nature" imply a causal relation? Would not be best to use another term such as overlap or co-ocurrence? Of course, I wouldI leave this up to the authors, just be careful not to mislead the reader. On a minor note I think the expression is too wordy, the word nature does not add meaning.

Introduction

Point 3. I would suggest the author to reformulate the hypotheses (rather than the posed hypothesis) in a clearer manner…The aims are clearly layed out, however the actual hypothesis is phrased in an unusual way and is a 3 lines long sentence. The authors formulate 3 aims, I think you should pose a hypothesis (or research question) for each of them.

Point 4. In the introduction section or in the measurement section I am missing more information about what gender norms are we talking about, I get that the test is based on US gender norms? DO the same norms apply in Spain? There are any Western gender norms that would be applicable to both? Could the authors please explain a bit more (Further that the succinctly mentioned Gender role theory in lines 73-77, and more linked to the norms understudy such as “domestic” or “relational”)

Methods

Point 5. Please provide more information about the assessment tool and try to maintain the same structure for both (you give anchors at one and the whole scale for the other). Please provide clear definitions and examples of each category if possible (as mentioned above, some categories I could figure, but I think some require an explanation, e.g. “domestic” or “relational”).

Point 6. Forty minutes seems like and awfully long time for 65 items (45+20). Maybe I did not get how long the CDA works, it is made up of 20 items per scale or as a whole…If it is 20 items I think it is too much time for university students.

Point 7. On line 157-158 you state “Reliability in the Spanish population yielded a Cronbach’s alpha coefficient of .76 for the total scale.” Does this alpha refer to your sample or the general population, which population? Related to this, does it make sense to present a global alpha for the whole scale or would it make more sense to present an alpha value for each scale?

Results

Point 8. In table 1 I do not understand the second line. Should I understand that 9.5% of the sample was Hetero-LGB? What does Hetero-LGB stand for?

Point 9. If all the CDA items start at 1 (never, line 146)…how can the range be between 0 and 6 for control and direct victimization in table 1?

Looking at the means I understand that mean of scales was used for the CDA (range 1-4), however total score was used for feminity (range 0-15)? Why that difference?

Point 10. I did not find how the prevalence was estimated. Scores equal or above 2? So if a girl is not controlling but had in the past year (response 2) controlled the time that her partner lasted connected once or twice (response 3) could you consider that Cyber dating abuse perpetration? Is that the usual criterion for CDA or is it provided by the original authors?

Point 11. In the regression analyses I do not understand which variables were used. On table 3, the DV is cyberdating victimization and direct perpetration is a predictor, but then in table 4 the DV is perpetration and control victimization is the predictor. Why do the authors not use control perpetration for the victimization prediction and the same goes for direct victimization and perpetration. I am missing something?

Discussion

Point 12. In line In line 231 I would advise the authors to be careful with implying causality, “female perpetrators tends (sic) to become victims” is a tricky sentence, that implies a temporal causality

Point 13. I thank the authors for the comprehensive number of limitations that they state. I think it is rather unusual for authors to acknowledge all on them some well and constructively . Thank you for the honesty.

Line 269 avoid the use of the word recent. It is (quite) recent, and hopefully the article will be published soon (2019, two years already) but it will not be recent in a few years time.

Minor mistakes or typos

Line 58 explainED

Line 72 research should ANALYZE (omit the D) differences

Line 185 last we(?) months        

Table 3’s title has two periods

Participants is point 2.2 (not 22.2)

When talking about reliability some of the alphas are preceded by a 0 (and once by a double zero) and sometimes by a point. I think since the alpha value can only reach 1, that you can present the number as .XX…any way, be consistent with it.

Author Response

Point 1. In the abstract the authors state

Indeed, the hierarchical regression analyses revealed a close association between perpetration and victimization behaviors in both direct and control forms of abuse examined.

However I am not sure if both forms of perpetration and victimization were included in the analysis, please see the point 11 in the results section.

Author’s answer: The information offered in the abstract is correct. All the forms were included in the analysis. However, we only included a table for those forms where (although small) significant effect were found about the relationships of feminine gender norms and cyber dating abuse.

In the case of the victimization, Table 3 includes data related to direct cyber dating victimization (where significant effects were found in relation to gender norms). The information regarding control victimization and the associations with control perpetration were pointed in the text (page 7, line 601-t04).

In the case of the perpetration, Table 4 include data concerning control perpetration (where significant effects were found in relation to gender norms). The information regarding direct perpetration was include in the text (page 7, line 606-609).

Point 2. I find myself wondering, does the first aim "explore the bidirectionality nature" imply a causal relation? Would not be best to use another term such as overlap or co-ocurrence? Of course, I would leave this up to the authors, just be careful not to mislead the reader. On a minor note I think the expression is too wordy, the word nature does not add meaning.

Author’s answer: you are right. Thank you for your recommendation. Given the cross-sectional nature of the study we have to avoid causality and we opted to use the term co-occurrence, and overlap to avoid reiteration of the terms.

Point 3. I would suggest the author to reformulate the hypotheses (rather than the posed hypothesis) in a clearer manner…The aims are clearly layed out, however the actual hypothesis is phrased in an unusual way and is a 3 lines long sentence. The authors formulate 3 aims, I think you should pose a hypothesis (or research question) for each of them.

Authors’ answer: thank you for your suggestion. We have included a subsection named “the present study” and we have included a hypothesis for each objective.

Point 4. In the introduction section or in the measurement section I am missing more information about what gender norms are we talking about, I get that the test is based on US gender norms? DO the same norms apply in Spain? There are any Western gender norms that would be applicable to both? Could the authors please explain a bit more (Further that the succinctly mentioned Gender role theory in lines 73-77, and more linked to the norms understudy such as “domestic” or “relational”)

Authors’ answer: We have now developed more carefully the gender role theory in the introduction (page 2, lines 123- 135). We have also discussed the similarities between USA and Spain regarding gender roles (page 3, lines 316-326). We have also explained the gender norms analyzed in the description of the instrument (page 4 and 5).

Methods

Point 5. Please provide more information about the assessment tool and try to maintain the same structure for both (you give anchors at one and the whole scale for the other). Please provide clear definitions and examples of each category if possible (as mentioned above, some categories I could figure, but I think some require an explanation, e.g. “domestic” or “relational”).

Authors’ answer: thank you, following your suggestion more information has been added to the description of each subscale and reliability data has been provide for each subscale.

Point 6. Forty minutes seems like and awfully long time for 65 items (45+20). Maybe I did not get how long the CDA works, it is made up of 20 items per scale or as a whole…If it is 20 items I think it is too much time for university students.

Author’s answer: thanks for your comment. It was a mistake. This study was part of a larger project and we indicated the whole time employed to fill other instruments. The average time was 15 minutes for the three scales.

Point 7. On line 157-158 you state “Reliability in the Spanish population yielded a Cronbach’s alpha coefficient of .76 for the total scale.” Does this alpha refer to your sample or the general population, which population? Related to this, does it make sense to present a global alpha for the whole scale or would it make more sense to present an alpha value for each scale?

Author’s answer: thank you for your comment. It was also a mistake, we mean “the current sample”. We have maintained the total alpha and included the alpha for each subscale as also suggested in Point 5.

Results

Point 8. In table 1 I do not understand the second line. Should I understand that 9.5% of the sample was Hetero-LGB? What does Hetero-LGB stand for?

Author’s answer: thank you for your comment. We agree that it could be confusing so we have decided to delete it. We tried to indicate that we had two categories regarding sexual orientation: 1) heterosexual and 2) LGB community.

Point 9. If all the CDA items start at 1 (never, line 146)…how can the range be between 0 and 6 for control and direct victimization in table 1?

Author’s answer: thank you for your comment. It was a typo. We have modified data in the table and indicated that the range was between 1 and 6.

Looking at the means I understand that mean of scales was used for the CDA (range 1-4), however total score was used for femininity (range 0-15)? Why that difference?

Author’s answer: well point. We have worked with total scores in the CFNI because it is the data returned by colleagues of the authors. Thus, when you have permission to use the CFNI you have to return the data gathered to the authors of the scale (or colleagues) in order to obtain scores of the subscales. They work with total scores instead of mean values, and that is the reason to work with total scores.

Point 10. I did not find how the prevalence was estimated. Scores equal or above 2? So if a girl is not controlling but had in the past year (response 2) controlled the time that her partner lasted connected once or twice (response 3) could you consider that Cyber dating abuse perpetration? Is that the usual criterion for CDA or is it provided by the original authors?

Authors’ answer: thank you for your comment. We followed the same produced used by the authors of the scale used (two or more times on at least three of the behaviors assessed for each role). We have included this information in the analyses section.

Point 11. In the regression analyses I do not understand which variables were used. On table 3, the DV is cyber dating victimization and direct perpetration is a predictor, but then in table 4 the DV is perpetration and control victimization is the predictor. Why do the authors not use control perpetration for the victimization prediction and the same goes for direct victimization and perpetration. I am missing something?

Authors’ answer: thank you for your comment. As we explained in point 1 the information about some of the relationships were not included in tables given the absence of significant relationships between cyber dating abuse and gender norms. However, significant relationships were found between victimization and perpetration in both forms of abuse. That data is reported in the text.

Discussion

Point 12. In line In line 231 I would advise the authors to be careful with implying causality, “female perpetrators tends (sic) to become victims” is a tricky sentence, that implies a temporal causality

Authors’ answer: thank you. You are right. We have indicated that “they are more likely to…” (line 267-268)

Point 13. I thank the authors for the comprehensive number of limitations that they state. I think it is rather unusual for authors to acknowledge all on them some well and constructively . Thank you for the honesty.

Authors’ answer: thank you for your kind comments and your comprehension.

Line 269 avoid the use of the word recent. It is (quite) recent, and hopefully the article will be published soon (2019, two years already) but it will not be recent in a few years time.

Authors’ answer: thank you for your suggestions. We have eliminated the word “recent”.

Minor mistakes or typos

Line 58 explainED

Line 72 research should ANALYZE (omit the D) differences

Line 185 last we(?) months        

Table 3’s title has two periods

Participants is point 2.2 (not 22.2)

When talking about reliability some of the alphas are preceded by a 0 (and once by a double zero) and sometimes by a point. I think since the alpha value can only reach 1, that you can present the number as .XX…any way, be consistent with it.

Authors’ answer: thank you for your careful reading.

Reviewer 2 Report

This study examines the bidirectionality nature of cyber dating abuse and its associations among conformity to feminine gender norms and cyber dating abuse.

As far as the introduction, the beginning sentence should capture the readers’ attention more readily. Consider adding a significant statistic, or something that captures your point in a more interesting way.

When discussing transmission of violence theory’s theorem try to provide an example to help readers understand the theory. Some readers may be unfamiliar with this theory, so provide more information about it. Also, cite the authors’ developers of the theory. (Lines 58-62)

Under the analysis plan, please include the software you used to analyze the data. Also, how was the data prepared/cleaned? How did you handle missing values?

The analyses were performed effectively and the tables are also quite clear. However, check with APA on developing tables, on the bottom of the tables you should provide basic descriptions of what the symbols mean.

The discussion provides an extensive discussion of the findings, but I would like you to bring forth the two theories that you began talking about in your literature review. How do these findings extend the theory, or how did the theory help to make interpretations of the findings of this study?

You do a fine job at discussion the limitations and the future directions of this study.

Overall, the manuscript has great potential and it is one of the few that examines this issue effectively. Best of luck as you continue to work on this manuscript.

Author Response

As far as the introduction, the beginning sentence should capture the readers’ attention more readily. Consider adding a significant statistic, or something that captures your point in a more interesting way.

Authors’ answer: thank you for your suggestion. We have modify the beginning paragraph following your advice (page 1, lines 25-32)

When discussing transmission of violence theory’s theorem try to provide an example to help readers understand the theory. Some readers may be unfamiliar with this theory, so provide more information about it. Also, cite the authors’ developers of the theory. (Lines 58-62)

Authors’ answer: We apologize for that. We were referring to the Social Learning Theory that include the idea of the intergenerational transmission of violence. We have now develop more carefully that paragraph (page 2, lines 105-116)

Under the analysis plan, please include the software you used to analyze the data. Also, how was the data prepared/cleaned? How did you handle missing values?

Authors’ answer: thank you for your suggestion. Software used has now been included in the analysis section. Preparation of data has been explained in the description of the participants.

The analyses were performed effectively and the tables are also quite clear. However, check with APA on developing tables, on the bottom of the tables you should provide basic descriptions of what the symbols mean.

Authors’ answer: thank you. We have now included the description of the symbols included in the tables.

The discussion provides an extensive discussion of the findings, but I would like you to bring forth the two theories that you began talking about in your literature review. How do these findings extend the theory, or how did the theory help to make interpretations of the findings of this study?

Authors’ answer: Thank you for your suggestion. We have now included the theory in the discussion and how it help to understand our results (page 8, lines 632-648),

Overall, the manuscript has great potential and it is one of the few that examines this issue effectively. Best of luck as you continue to work on this manuscript.

Authors’ answer: thank you for you helpful and kind review.

Round 2

Reviewer 1 Report

After reviewing the new version of the manuscript and the authors' responses, I consider the manuscript suitable for publication .It makes a sound and valuable contribution that will be useful for the journal’s readership and academics.  

I can only congratulate the authors for their excellent work.  They have responded to all the comments I made in a more than adequate manner and both the review and the manuscript demonstrates a high degree of scientific and academic rigour. The objectives and hypotheses are organized in a clearer manner, the instruments are described in more detail, the participant and procedure section provide the necessary information allowing for replicability and both the introduction and the discussion have been enhanced with new and appropriate information.

I look forward to seeing it published.

P.s: Please do no to forget to delete the unnecessary marking lines in Table 1

Author Response

Thank you for your kind comments. We have deleted unnecessary lines in Table 1.

Best.